# Learning Approximate Distribution-Sensitive Data Structures

**Zenna Tavares**
MIT
zenna@mit.edu

**Armando Solar Lezama**
MIT
asolar@csail.mit.edu

## Abstract

We model representations as data-structures which are distribution sensitive, i.e., which exploit regularities in their usage patterns to reduce time or space complexity. We introduce probabilistic axiomatic specifications to extend abstract data structures - which specify a class of representations with equivalent logical behavior - to a distribution-sensitive data structures. We reformulate synthesis of distribution-sensitive data structures as a continuous function approximation problem, such that the functions of a data-structure deep neural networks, such as a stack, queue, natural number, set, and binary tree.

## 1 Introduction

Recent progress in artificial intelligence is driven by the ability to learn representations from data. Yet not all kinds of representations are equal, and many of the fundamental properties of representations (both as theoretical constructs and as observed experimentally in humans) are missing. Perhaps the most critical property of a system of representations is compositionality, which as described succinctly in (Fodor & Lepore, 2002), is when (i) it contains both primitive symbols and symbols that are complex; and (ii) the latter inherit their syntactic/semantic properties from the former. Compositionality is powerful because it enables a system of representation to support an infinite number of semantically distinct representations by means of combination. This argument has been supported experimentally; a growing body of evidence (Spelke & Kinzler, 2007) has shown that humans possess a small number of primitive systems of mental representation - of objects, agents, number and geometry - and new representations are built upon these core foundations.

Representations learned with modern machine learning methods possess few or none of these properties, which is a severe impediment. For illustration consider that navigation depends upon some representation of geometry, and yet recent advances such as end-to-end autonomous driving (Bojarski et al., 2016) side-step building explicit geometric representations of the world by learning to map directly from image inputs to motor commands. Any representation of geometry is implicit, and has the advantage that it is economical in only possessing information necessary for the task. However, this form of representation lacks (i) the ability to reuse these representations for other related tasks such as predicting object stability or performing mental rotation, (ii) the ability to compose these representations with others, for instance to represent a set or count of geometric objects, and (iii) the ability to perform explicit inference using representations, for instance to infer why a particular route would be faster or slower.

This contribution provides a computational model of mental representation which inherits the compositional and productivity advantages of symbolic representations, and the data-driven and economical advantages of representations learned using deep learning methods. To this end, we model mental representations as a form of data-structure, which by design possess various forms of compositionality. In addition, in step with deep learning methods we refrain from imposing a particular representations on a system and allow it instead be learned. That is, rather than specify a concrete data type (for example polygons or voxels for geometry), we instead define a class of representations as abstract data types, and impose invariants, or axioms, that *any* representation must adhere to.

Mathematicians have sought an axiomatic account of our mental representations since the end of the nineteenth century, but both as an account of human mental representations, and as a means of specifying representations for intelligent systems, the axiomatic specifications suffer from a number

of problems. Axioms are universally quantified - for *all* numbers, sets, points, etc - while humans, in contrast, are not uniformly good at manipulating numbers of different magnitude (Hyde, 2011; Nuerk & Willmes, 2005; Dehaene, 1997), rotating geometry of different shapes (Izard et al., 2011), or sets of different cardinality. Second, axioms have no algorithmic content; they are declarative rules which do not suggest how to construct concrete representations that satisfy them. Third, only simple systems have reasonable axioms, whereas many representations are complex and cannot in practice be fully axiomitized; conventional axiomatic specifications do not readily accommodate partial specification. A fourth, potentially fatal threat is offered by Dehaene (1997), where he shows that there are infinitely many systems, most easily dismissed by even a child as clearly not number-like, which satisfy Peano's axioms of arithmetic. Moreover these "nonstandard models of arithmetic" can never be eliminated by adding more axioms, leading Dehaene to conclude "Hence, our brain does not rely on axioms.".

We extend, rather than abandon, the axiomatic approach to specifying mental representations, and employ it purely as mechanism to embed domain specific knowledge. We model a mental representation as an implementation of an abstract data type which adheres approximately to a *probabilistic* axiomatic specification. We refer to this implementation as a distribution-sensitive data-structure.

In summary, in this paper:

- We introduce probabilistic axiomatic specifications as a quantifier-free relaxation of a conventional specification, which replaces universally quantified variables with random variables.

- Synthesis of a representation is formulated as synthesis of functions which collectively satisfy the axioms. When the axioms are probabilistic, this is amounts of maximizing the probability that the axiom is true.

- We present a number of methods to approximate a probabilistic specification, reducing it to a continuous loss function.

- We employ neural networks as function approximators, and through gradient based optimization learn representations for a number of fundamental data structures.

## 2 BACKGROUND: ABSTRACT DATA TYPES

Abstract data types model representations as a set of types and functions which act on values of those types. They can also be regarded as a generalized approach to algebraic structures, such as lattices, groups, and rings. The prototypical example of an abstract data type is the $Stack$, which models an ordered, first-in, last-out container of items. We can abstractly define a $Stack$ of $Items$, in part, by defining the interface:

$$empty : Stack$$
$$push : Stack \times Item \to Stack$$
$$pop : Stack \to Stack \times Item$$
$$isempty : Stack \to \{0, 1\}$$

The interface lists the function names and types (domains and range). Note that this is a functional (rather than imperative) abstract data type, and each function in the interface has no internal state. For example, $push$ is a function that takes an instance of a Stack and an $Item$ and returns a $Stack$. $empty : Stack$ denotes a constant of type $Stack$, the empty stack of no items.

The meaning of the constants and functions is not specified in the interface. To give meaning to these names, we supplement the abstract data type with a specification as a set of axioms. The specification as a whole is the logical conjunction of this set of axioms. Continuing our example, for all $s \in Stack, i \in Item$:

$$pop(push(s, i)) = (s, i) \tag{1}$$
$$isempty(empty) = 1 \tag{2}$$
$$isempty(push(s, i)) = 0 \tag{3}$$
$$pop(empty) = \perp \tag{4}$$

A concrete representation of a stack is a data structure which assigns constants and functions to the names $empty$, $push$, $pop$ and $isempty$. The data structure is a stack if and only if it satisfies the specification.

## 2.1 Composition of data structures

There are a number of distinct forms of compositionality with respect to data structures. One example is algorithmic compositionality, by which we can compose algorithms which use as primitive operations the interfaces to these representations. These algorithms can in turn form the interfaces to other representations, and so on.

An important property of an abstract data types which supports algorithmic compositionality is encapsulation. Encapsulation means that the particular details of how the functions are implemented should not matter to the user of the data type, only that it behaves as specified. Many languages enforce that the internals are unobservable, and that the data type can only be interacted with through its interface. Encapsulation means that data-structures can be composed without reasoning about their internal behavior.

In this paper however, we focus on parametric compositionality. Some data structures, in particular containers such as a stack, or set, or tree are parametric with respect to some other type, e.g. the type of item. Parametric compositionality means for example that if we have a representation of a set, and a representation of a number, we get a set of numbers for free. Or, given a representations for a tree and representations for Boolean logic, we acquire the ability to form logical expressions for free.

## 2.2 Distribution Sensitive Data Structures

Axiomatic specifications almost always contain universal quantifiers. The stack axioms are quantified over all possible stacks and all possible items. Real world use of a data structure is however never exhaustive, and rarely uniform. Continuing our stack example, we will never store an infinite number of items, and the distribution over how many items are stored, and in which order relative to each other, will highly non-uniform in typical use cases. Conventional data structures are agnostic to these distributional properties.

Data structures that exploit non-uniform query distributions are typically termed *distribution-sensitive* (Bose et al., 2013), and are often motivated by practical concerns since queries observed in real-world applications are not uniformly random. An example is the optimum binary search tree on $n$ keys, introduced by Knuth (Bose et al., 2013), which given a probability for each key has an average search cost no larger than any other key. More generally, distribution-sensitive data structures exploit underlying patterns in a sequence of operations in order to reduce time and space complexity.

## 3 Probabilistic Axiomatic Specification

To make the concept of a distribution-sensitive data-structure precise, we first develop the concept of an probabilistically axiomatized abstract data type $(T, O, F)$, which replaces universally quantified variables in its specification with random variables. $T$ and $O$ are respectively sets of type and interface names. $F$ is a set of type specifications, each taking the form $m : \tau$ for a constant of type $\tau$, or $o : \tau_1 \to \tau_2$ denoting a function from $\tau_1$ to $\tau_2$. Here $\tau \in T$ or a Cartesian product $T_1 \times \cdots \times T_n$.

A concrete data type $\sigma$ implements an abstract data type by assigning a value (function or constant) to each name in $O$. A concrete data type is deemed a valid implementation only with respect to an algebraic specification $A$. $A$ is a set of equational axioms of the form $p = q$, $p$ and $q$ are constants, random variables, or transformations of random variables by functions in $O$.

Since a transformation of a random variable yields a random variable, and an axiom is simply a predicate of its left and right hand side arguments, random variables present in an axiom implies that the axiom itself is a Boolean valued random variable. For example if we have a distribution over items $i$ of the stack, axiom (1) itself is a random variable which is true or false depending on $i$, $push$, $pop$, and can only be satisfied with some probability. We let $P[A(\sigma)]$ denote the probability

that a concrete data type $\sigma$ satisfies the axioms:

$$P[A(\sigma)] \coloneqq P[\wedge_i p_i = q_i] \qquad (5)$$

Probabilistic axioms do not imply that the concrete data-structure itself is probabilistic. On the contrary, we are concerned with specifying and synthesizing deterministic concrete data structures which exploit uncertainty stemming only from the patterns in which the data-structure is used.

When $P[A(\sigma)] = 1$, $\sigma$ can be said to fully satisfy the axioms. More generally, with respect to a space $\Sigma$ of concrete data types, we denote the maximum likelihood $\sigma^*$ as one which maximizes the probability that the axioms hold:

$$\sigma^* = \arg\max_{\sigma \in \Sigma} P[A(\sigma)] \qquad (6)$$

## 4 Approximate Abstract Data Types

A probabilistic specification is not easier to satisfy than a universally quantified one, but it can lend itself more naturally to a number of approximations. In this section we outline a number of relaxations we apply to a probabilistic abstract data type to make synthesis tractable.

### Restrict types to real valued arrays

Each type $\tau \in T$ will correspond to a finite dimensional real valued multidimensional array $\mathbb{R}^n$. Interface functions are continuous mappings between these arrays.

### Unroll Axioms

Axiom (1) of the stack is intensional in the sense that it refers to the underlying stack $s$. This provides an inductive property allowing us to fully describe the behavior of an unbounded number of $push$ and $pop$ operations with a single equational axiom. However, from an extensional perspective, we do not care about the internal properties of the stack; only that it behaves in the desired way. Put plainly, we only care that if we push an item $i$ to the stack, then pop, that we get back $i$. We do not care that the stack is returned to its initial state, only that it is returned to some state that will continue to obey this desired behavior.

An extensional view leads more readily to approximation; since we cannot expect to implement a stack which satisfies the inductive property of axiom 1 if it is internally a finite dimensional vector. Instead we can unroll the axiom to be able to stack some finite number of $n$ items:

### Approximate Distributions with Data

We approximate random variables by a finite data distribution assumed to be a representative set of samples from that distribution. Given an axiom $p = q$, we denote $\hat{p}$ and $\hat{q}$ as values (arrays) computed by evaluating $p$ and $q$ respectively with concrete data from the data distributions of random variables and the interface functions.

### Relax Equality to Distance

We relax equality constraints in axioms to a distance function, in particular the L2 norm. This transforms the equational axioms into a loss function. Given $i$ axioms, the approximate maximum likelihood concrete data type $\hat{\sigma}^*$ is then:

$$\hat{\sigma}^* = \arg\min_i \sum \|\hat{p}_i - \hat{q}_i\| \qquad (7)$$

Constants and parameterized functions (e.g. neural networks) which minimizes this loss function then compose a distribution-sensitive concrete data type.

## 5 Experiments

We successfully synthesized approximate distribution-sensitive data-structures from a number of abstract data types:

- Natural number (from Peano's axioms)
- Stack
- Queue
- Set
- Binary tree

With the exception of natural number (for which we used Peano's axioms), we use axiomitizations from (Dale & Walker, 1996). As described in section 4, since we use finite dimensional representations we unroll the axioms some finite number of times (e.g., to learn a stack of three items rather than it be unbounded) and "extensionalize" them.

In each example we used we used single layer convolutional neural networks with 24, 3 by 3 filters and rectifier non-linearities. In container examples such as $Stack$ and $Queue$, the $Item$ type was sampled from MNIST dataset, and the internal stack representation was chosen (for visualization) to also be a 28 by 28 matrix. We minimized the equational distance loss function described in section 3 using the adam optimization algorithm, with a learning rate of 0.0001 In figures 1 and 2 we visualize the properties of the learned stack.

To explore compositionality, we also learned a $Stack$, $Queue$ and $Set$ of $Number$, where $Number$ was itself a data type learned from Peano's axioms.

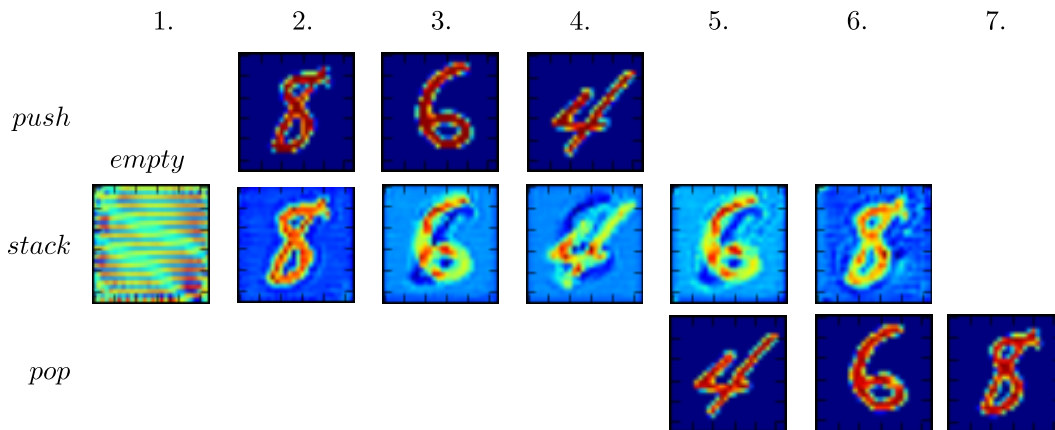

Figure 1: Validation of stack trained on MNIST digits, and introspection of internal representation. Row $push$ shows images pushed onto stack from data in sequence. Row $pop$ shows images taken from stack using $pop$ function. Their equivalence demonstrates that the stack is operating correctly. Row $stack$ shows internal representation after $push$ and $pop$ operations. The stack is represented as an image of the same dimension as MNIST (28 by 28) arbitrarily. The stack learns to compress three images into the the space of one, while maintaining the order. It deploys an interesting interlacing strategy, which appears to exploit some derivative information.

## 6 Analysis

The learned internal representations depend on three things (i) the axioms themselves, (ii) the architecture of the networks for each function in the interface, and (iii) the optimization procedure. In the stack example, we observed that if we decreased the size of the internal representation of a stack, we would need to increase the size and complexity of the neural network to compensate. This implies that statistical information about images must be stored somewhere, but there is some flexibility over where.

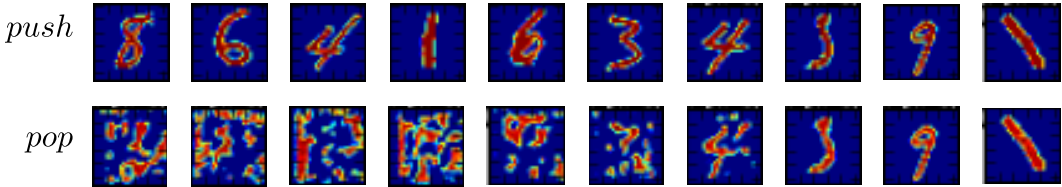

Figure 2: Generalization of the stack. Top left to top right, 10 images stacked in sequence using *push*. Bottom right to bottom left: result from calling *pop* on stack 10 times. This stack was trained to stack three digits. It appears to generalize partially to four digits but quickly degrades after that. Since the stack is finite dimensional, it is not possible for it to generalize to arbitrarily long sequences of *push* operations.

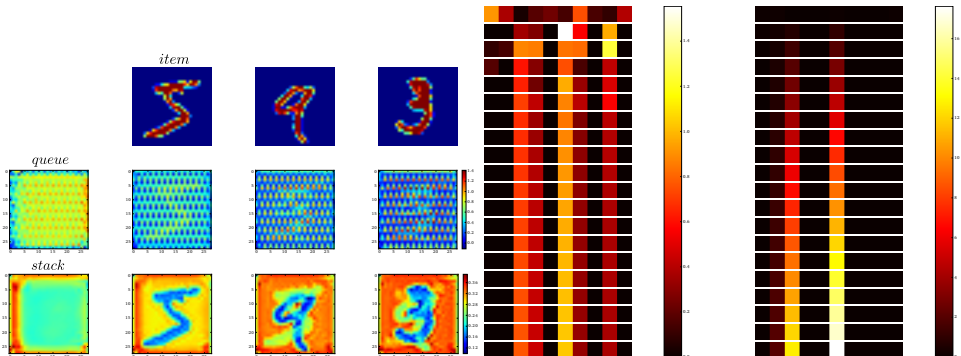

Figure 3: Left: Stack versus queue encoding. Three MNIST images (top row) were enqueued onto the empty queue (middle row left), and pushed onto the empty stack (bottom row left). Middle row shows the internal queue representation after each *enqueue* operation, while bottom is the internal stack representation after each *push*. In this case, the learned stack representation compresses pixel intensities into different striated sections of real line, putting data about the first stacked items at lower values and then shifting these to higher values as more items are stacked. This strategy appears different from that in figure 1, which notably was trained to a lower error value. The internal queue representation is less clear; the hexagonal dot pattern may be an artifact of optimization or critical to its encoding. Both *enqueue* and *push* had the same convolutional architecture. Right: Internal representations of natural numbers from 0 (top) to 19 (bottom). Natural numbers are internally represented as a vector of 10 elements. Number representations on the left are found by repeateding the succesor function, e.g. ($succ(zero)$, $succ(succ(zero))$, ...). Numbers on the right are found by encoding machine integers into this internal representation.

Given the same architecture, the system learned different representations depending on the axioms and optimization. The stack representation learned in figure 1 differs from that in figure 3, indicating that there is not a unique solution to the problem, and different initialization strategies will yield different results. The queue internal representation is also different to them both, and the encoding is less clear. The queue and stack representations could have been the same (with only the interface functions *push*, *pop*, *queue* and *dequeue* taking different form).

As shown in figure 2, data-structures exhibit some generalization beyond the data distributions on which they are trained. In this case, a stack trained to store three items, is able to store four with some error, but degrades rapidly beyond that. Of course we cannot expect a finite capacity representation to store an unbounded number of items; lack of generalization is the cost of having optimized performance on the distribution of interest.

## 7 RELATED WORK

Our contribution builds upon the foundations of distribution-sensitive data structures (Bose et al., 2013), but departs from conventional work on distribution-sensitive data structures in that: (i) we

synthesize data structures automatically from specification, and (ii) the distributions of interest are complex data distributions, which prevents closed form solutions as in the optimum binary tree.

Various forms of machine learning and inference learn representations of data. Our approach bears resemblance to the auto-encoder (Bengio, 2009), which exploits statistics of a data distribution to learn a compressed representation as a hidden layer of a neural network. As in our approach, an auto-encoder is distribution sensitive by the constraints of the architecture and the training procedure (the hidden layer is of smaller capacity than the data and which forces the exploitation of regularities). However, an auto-encoder permits just two operations: encode and decode, and has no notion explicit notion of compositionality.

A step closer to our approach than the auto-encoder are distributed representations of words as developed in (Mikolov et al., 2000). These representations have a form of compositionality such that vector arithmetic on the representation results in plausible combinations (Air + Canada = Air-Canada).

Our approach to learning representation can be viewed as a form of data-type synthesis from specification. From the very introduction of abstract data types, verification that a given implementation satisfies its specification was a motivating concern (Guttag et al., 1978; Guttag, 1978; Spitzen & Wegbreit, 1975). Modern forms of function synthesis (Solar-Lezama, 2009; Polikarpova & Solar-Lezama, 2016) use verification as an oracle to assist with synthesis. Our approach in a broad sense is similar, in that derivatives from loss function which is derived from relaxing the specification, guide the optimization through the paramterized function spaces.

Probabilistic assertions appear in first-order lifting (Poole, 2003), and Sampson (Sampson et al., 2014) introduce probabilistic assertions. Implementation of data type is a program. Main difference is that we synthesize data type from probabilistic assertion. Sumit's work (Sankaranarayanan, 2014) seeks upper and lower bounds for the probability of the assertion for the programs which operate on uncertain data.

Recent work in deep learning has sought to embed discrete data structures into continuous form. Examples are the push down automata (Sun et al., 1993), networks containing stacks (Grefenstette et al., 2015), and memory networks (Sukhbaatar et al., 2015). Our approach can be used to synthesize arbitrary data-structure, purely from its specification, but is parameterized by the neural network structure. This permits it more generality, with a loss of efficiency.

## 8  DISCUSSION

In this contribution we presented a model of mental representations as distribution sensitive data structures, and a method which employs neural networks (or any parameterized function) to synthesize concrete data types from a relaxed specification. We demonstrated this on a number of examples, and visualized the results from the stack and queue.

One of the important properties of conventional data structures is that they compose; they can be combined to form more complex data structures. In this paper we explored a simple form of parametric composition by synthesizing containers of numbers. This extends naturally to containers of containers, .e.g sets of sets, or sets of sets of numbers. Future work is to extend this to richer forms of composition. In conventional programming languages, trees and sets are often made by composing arrays, which are indexed with numbers. This kind of composition ls fundamental to building complex software from simple parts.

In this work we learned representations from axioms. Humans, in contrast, learn representations mostly from experience in the world. One rich area of future work is to extend data-structure learning to the unsupervised setting, such that for example an agent operating in the real world would learn a geometric data-structures purely from observation.

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
