# Peer review of "Learning Approximate Distribution-Sensitive Data Structures"

_ICLR 2017 — rejected_

[Official Review · AnonReviewer3 · rating 4 · confidence 3 · 16 Dec 2016]
**Interesting direction, but not there yet.**

A method for training neural networks to mimic abstract data structures is presented. The idea of training a network to satisfy an abstract interface is very interesting and promising, but empirical support is currently too weak. The paper would be significantly strengthened if the method could be shown to be useful in a realistic application, or be shown to work better than standard RNN approaches on algorithmic learning tasks.

The claims about mental representations are not well supported. I would remove the references to mind and brain, as well as the more philosophical points, or write a paper that really emphasizes one of these aspects and supports the claims.

[Official Review · AnonReviewer1 · rating 4 · confidence 3 · 16 Dec 2016]
**No Title**

The paper presents a framework to formulate data-structures in a learnable way. It is an interesting and novel approach that could generalize well to interesting datastructures and algorithms. In its current state (Revision of Dec. 9th), there are two strong weaknesses remaining: analysis of related work, and experimental evidence.

Reviewer 2 detailed some of the related work already, and especially DeepMind (which I am not affiliated with) presented some interesting and highly related results with its neural touring machine and following work. While it may be of course very hard to make direct comparisons in the experimental section due to complexity of the re-implementation, it would at least be very important to mention and compare to these works conceptually.

The experimental section shows mostly qualitative results, that do not (fully) conclusively treat the topic. Some suggestions for improvements:
* It would be highly interesting to learn about the accuracy of the stack and queue structures, for increasing numbers of elements to store.
* Can a queue / stack be used in arbitrary situations of push-pop operations occuring, even though it was only trained solely with consecutive pushes / consecutive pops? Does it in this enhanced setting `diverge' at some point?
* The encoded elements from MNIST, even though in a 28x28 (binary?) space, are elements of a ten-element set, and can hence be encoded a lot more efficiently just by `parsing' them, which CNNs can do quite well. Is the NN `just' learning to do that? If so, its performance can be expected to strongly degrade when having to learn to stack more than 28*28/4=196 numbers (in case of an optimal parser and loss-less encoding). To argue more in this direction, experiments would be needed with an increasing number of stack / queue elements. Experimenting with an MNIST parsing NN in front of the actual stack/queue network could help strengthening or falsifying the claim.
* The claims about `mental representations' have very little support throughout the paper. If indication for correspondence to mental models, etc., could be found, it would allow to hold the claim. Otherwise, I would remove it from the paper and focus on the NN aspects and maybe mention mental models as motivation.

[Official Review · AnonReviewer2 · rating 3 · confidence 4 · 16 Dec 2016]

The paper presents a way to "learn" approximate data structures. They train neural networks (ConvNets here) to perform as an approximate abstract data structure by having an L2 loss (for the unrolled NN) on respecting the axioms of the data structure they want the NN to learn. E.g. you NN.push(8), NN.push(6), NN.push(4), the loss is proportional to the distance with what is NN.pop()ed three times and 4, 6, 8 (this example is the one of Figure 1).

There are several flaws:
 - In the case of the stack: I do not see a difference between this and a seq-to-seq RNN trained with e.g. 8, 6, 4 as input sequence, to predict 4, 6, 8.
 - While some of the previous work is adequately cited, there is an important body of previous work (some from the 90s) on learning Peano's axioms, stacks, queues, etc. that is not cited nor compared to. For instance [Das et al. 1992], [Wiles & Elman 1995], and more recently [Graves et al. 2014], [Joulin & Mikolov 2015], [Kaiser & Sutskever 2016]...
 - Using MNIST digits, and not e.g. a categorical distribution on numbers, is adding complexity for no reason.
 - (Probably the biggest flaw) The experimental section is too weak to support the claims. The figures are adequate, but there is no comparison to anything. There is also no description nor attempt to quantify a form of "success rate" of learning such data structures, for instance w.r.t the number of examples, or w.r.t to the size of the input sequences. The current version of the paper (December 9th 2016) provides, at best, anecdotal experimental evidence to support the claims of the rest of the paper.

While an interesting direction of research, I think that this paper is not experimentally sound enough for ICLR.

[Final Decision · Program Chairs · 06 Feb 2017]
**ICLR committee final decision**

The consensus of the reviewers, although their reviews where somewhat succinct, was that the paper proposes an interesting research direction by training neural networks to approximate datastructures by constraining them to (attempt to) respect the axioms of the structure, but is thin on the ground in terms of evaluation and comparison to existing work in the domain (both in terms of models and "standard" experiments"). The authors have not sought to defend their paper against the reviewers' critique, and thus I am happy to accept the consensus and reject the paper.